# Shifting elder-care practices in Chinese middle-class families

**Lu Wang** [1,2]*, **Rose Gilroy**[3], **Andrew Law**[3]

**1** School of Architecture and Art Design, HeBei University of Technology, Tianjin, China, **2** Key Laboratory of Healthy Human Settlements in Hebei Province, HeBei University of Technology, Tianjin, China, **3** School of Architecture, Planning & Landscape, Newcastle University, Newcastle upon Tyne, United Kingdom

* 2022055@hebut.edu.cn

**Data Availability Statement:** A) Our manuscript alone currently contains our minimal data set. It includes that authors theme and summarize records from participants' interviews. It also anonymizes and tables participants' peraonal information. S1 Appendix with manucript has

## Abstract

This paper explores how ideas and practices of elder-care may be changing for the new Chinese middle class. This paper draws on in-depth interviews with the members of different generations of ten middle class families in the Chinese city of Tianjin. It explores how increased resources but also increased pressure are affecting the care of older people and the expectations around elder-care. Thematic analysis of transcripts entered into NVIVO revealed three findings. Firstly, that the generation born in the 1950s and 1960s are often negotiating their care responsibilities between their parents and their grandchildren. Secondly, the only viable elder-care solution for many families is to buy in support from an unregulated market-based home care sector. Thirdly driven by an increasingly postmodern culture, filial piety may be changing from a normative expectation to a set of new practices based on familial reciprocity. The paper concludes by reflecting on the issues that will arise from the proposed raising of the retirement age in China, and the increasing geographic dispersal of generations. Flexible working policies as well as investment in a regulated home care sector are recommended as solutions to be explored.

## Introduction change and tradition in China

The story of China since the mid-20th century is one of multiple and profound social and economic change. The population has aged in line with global trends but has been heavily impacted by the imposition of the one child policy in the early reform era of 1980. The dismantling of urban state-owned enterprises that were a conduit for cradle to grave support has led to Chinese families, like many families in other parts of the world, having to organise and meet the cost of care themselves. The change in economic policy that resulted in China opening up to the global economy has created new employment opportunities, which has laid greater emphasis on high educational achievement and has driven the growth of a new middle class. While much has changed, the ancient tradition of filial piety that demands care and support for older family members persists. How do these changes and traditions coexist? This paper explores the dilemmas facing middle class families and asks how they balance their resources across the multiple demands of the oldest and youngest generations. How do these societal changes affect and influence the expectations of different generations? Finally, this paper asks

shown the interview guidelines. The contents in this manuscript are with the consent of the participants.

**Funding:** The author(s) received no specific funding for this work.

**Competing interests:** Enter: The authors have declared that no competing interests exist.

what role, government and the market can play in supporting middle class families in meeting their care responsibilities.

## Understanding the middle class in China

In a period of less than 25 years, ideas of class in China have shifted from a Maoist labelling of peasants, landlords and the bourgeoisie to one that is based upon notions of economic status, education, and lifestyles. While entry into the middle class is defined in China by income bands [1], the experience of being middle class is more fittingly viewed through Bourdieu's [2] theories of the significance of capitals. Donald and Zheng [3] discuss the evidence of conscious identity building through middle class consumption. In a consumerist postmodern culture, there is an emphasis on personal choice. To what extent, therefore, does following traditional values and roles also become a matter of choice rather than a necessary discipline of social life [4]? In the context of an ageing society is the practice of filial piety in China simply another choice that can be either embraced or rejected.

## Challenges to family practices and filial piety

The concept of filial piety is based on a collection of philosophical ideas developed by the Chinese philosopher Confucius (*Kong Fuzi*, 551–479 BCE). Confucius stated that: 'In serving his parents, a filial son reveres them in daily life; he makes them happy when he nourishes them; he takes anxious care of them in sickness; he shows great sorrow over their death; and he sacrifices to them with solemnity' [5, 6]. The use of the pronoun 'he' is not merely an issue of semantics but reflects the importance of the father-son relationship since it is the son who continues the family line in a patriarchal society. To what extent does the greater equality of men and women and daughter only families challenge the notion of who takes responsibility in the Chinese family?

While a traditional value, the idea of filial piety has come under increasing scrutiny in modern China. The Maoist era established cradle to grave care for the urban Chinese through work units. At the same time, the allocation of housing was to couples with children creating a new emphasis on the importance of the nuclear family [7]. Following the demise of the work unit system, responsibilities for supporting older people and children were returned once again to the family [8]. However, the focus on the nuclear family was further emphasised by economic reform and progress toward a more 'modern society' that promotes 'conjugal' patterns of family life; accordingly, in this situation couples make decisions about their own lives and the lives of their young children with little influence or interference from their parents or other extended family members [9]. In reflecting on these notions, we might ask how these new ideas co-exist with the authority traditionally vested in older people by age?

The dynamics of family life were profoundly affected by the one-child policy of 1980. Although the one-child policy was substituted by a two-child policy in 2015 and a three-child policy in 2021, nevertheless, the original policy has shaped at least one generation of Chinese people who have no siblings. Today, the '4 grandparents-2- parents and 1 child' family is the norm in urban areas [10–12]. This has given rise to the growing importance of the child in the Chinese household but has also exposed the, as yet under-developed response to elder-care.

Because of the needs of the elder generation and the youngest, there is now competition for family resources [13]. A new phrase "the sandwich generation" has been coined to describe the stress of those sandwiched between their ageing parents and their own maturing children who have become child-rearers [14]. This generation needs to balance their resource flows of money, time, and people. Raising the younger generation is more expensive because of the increasing cost of education, housing, and marriage; while for the oldest family members, the

cost of medical care is continually rising, along with the cost of care homes. There are of course not just financial costs involved in this care, but also time and emotional investment [15]. Research by Wu et al. [16] reveals that the family resources of time and finance now tend to flow to the youngest. This has led some commentators, such as Ikels [17], to suggest that as Chinese people age, for the first time, they may not be able to count on their children for support. This paper poses questions about the dynamism of expectations surrounding shifting elder-care practices in Chinese middle-class families.

## Methodology

Research on family care practices in China is limited and therefore it became clear that a grounded theory approach was needed that would allow the concepts to arise from the data collected. The desire to explore these issues across generations also demanded a family-based approach using qualitative methods that would allow participants to take more control of the narratives. The use of a qualitative approach posed two complications. Firstly, research in China is dominated by quantitative methods that generate large numerical data sets and the first author was often met by respondents who were puzzled by the approach she was taking and the idea of qualitative work. Responding to this feedback the first author decided to insert a brief questionnaire into the research that asked for base line information about household composition, housing choices, contact with older and younger generations, and use of time and care preferences. Secondly gaining access to a family home is a privilege not easily granted. While one visit may be acceptable as an honoured guest, the request to repeat these visits and to ask about family practices was achieved only by working hard to establish trust. In part, the building of trust was also enabled by the decision to use a case study method and to undertake the field work in the first author's home city of Tianjin. The first author was a female doctoral candidate who had undertaken the post graduate research training offered as a part of her programme at Newcastle University.

### Sampling strategy

The need for trust made a random or scientific sampling framework ineffective for producing willing research participants. Instead, the first author used a two-pronged strategy by drawing on her own network of school friends and her parents' friendship circle. Secondly, the sample was enriched when the first author joined popular activity groups frequented by older people; in which, over time she became a more familiar figure. The need for the willingness of all generations to give consent led to several potential participants being rejected and a final total of ten families were selected for investigation where at least one of the generations could be defined as 'middle-class' in Chinese terms (Table 1).

These selection criteria were summarised and screened from the large corpus of Chinese statistical-survey research [1, 19, 20].

### Working in the field

The fieldwork was completed over a 6-month period from January to July 2019. For clarity, the oldest group of respondents are called G (generation) 1; their sons and daughters are labelled G2, and their single child is described as G3. Some families also had a G4, but no attempt was made to engage with young children. Sons and daughters are labelled as S and D and are ranked in birth order within their generation.

Interviews took place in the respective homes of the respondents. In conducting interviews with the respondents, a topic guide that drew on the family practices literature asked what later life meant to the participant; their familial relationships, views on filial piety; and their

**Table 1. Selection criteria for participants.**

| Criterion | Requirements |
|---|---|
| Annual income | Minimum 100K CNY per year (about £11,401). According to recent research middle-class household income of new Tier one city ranges from 200K-1500K CNY per year (£21,827-£163,705) [18]; 100K-960K CNY per year (£10,913-£104,771) according to World Bank. |
| Occupation | Specialist or skilled job, or managerial or comparable level, in either the state or private sector. |
| Education | Above bachelor's degree or college degree |
| Hukou (Residence Status) | Urban household registration |
| House/car | Own a house and/or car. The home may be mortgaged or owned outright. Resident in a gated community |
| Self-identity | Conscious of lifestyle and self-cultivation |

use of care resources. Several participants drew their family trees, which allowed the primary researcher to develop an easy-to-follow people resource map–an example of this map can be seen below in Fig 1, which presents the Wang family.

The dotted line indicates those who were interviewed, and the circle indicates the 'gate keeper or entry person' who arranged access to the family. Every family was visited at least twice and usually for a half day each time. During each stay, the researcher interviewed each generation separately for about 40 minutes. A total of 49 individuals participated.

Ethical approval for the research was gained through Newcastle University's postgraduate research ethics committee. In accordance with established ethical practice, participants received written information about the research and were asked to give written consent before taking part in the investigation. In the field participants were given letter and numerical identifiers and in subsequent publications they are given pseudonyms to preserve anonymity while retaining their individuality.

While the researcher's preference was to record the conversations, several participants—100% in G1 = 11; 96% in G2 = 24(total 25); 25% in G3 = 3 (total 13)—were hesitant and there were others who asked for the recording to be stopped after several minutes. On reflection, a number of reasons may explain this hesitancy: Generation 1 had memories of the Cultural Revolution (1966–1976) when a traceable record of one's views might be dangerous. The testimony of some G1 respondents demonstrated that experiences of that time were still felt deeply.

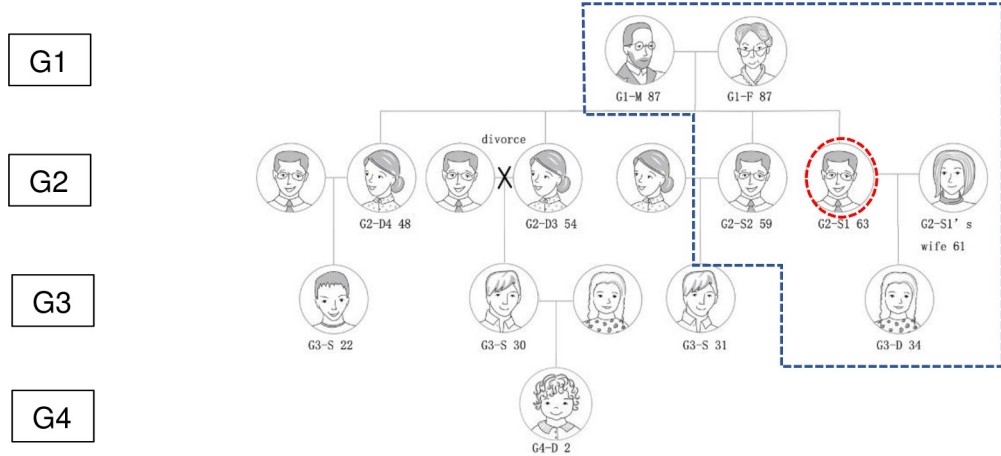

**Fig 1. The Wang family tree.**

Generation 2 often wanted to show the family's good side and were reluctant to be drawn into disclosing issues on tape that reflected poorly on their families (though off tape participants were critical of siblings and daughters). Generation 3 was content to be recorded but was reluctant to be photographed; indeed, as the research progressed, the investigator felt that this unwillingness might be because they were a highly image conscious generation. Data was recorded through note-taking and voice memos recorded immediately after the interviews along with extensive research diaries.

Each interview was transcribed by the first author, and interviewees were sent a copy of the transcript to review and edit if they so wished. An English language version was then stored in NVivo. The transcripts and research were shared with the second and third authors who discussed the framework for analysis.

## Data analysis

Drawing on a grounded theory approach, translated interviews were coded using NVivo with the researcher team collectively generating the headings for the coding process. Using the interview questions as an initial framework, but engaging in a process of critical reflection on the usefulness of the headings, generating sub-headings and new headings as close reading revealed new themes. The data was subject to several rounds of close reading until we were confident that no new themes were emerging. Once the coding was completed, narratives were grouped into thematic clusters of family strategies and generational issues shared or contested across families.

## Findings

Families were asked to detail their relationships, sources of income and wealth including property ownership (Table 2). Unlike some societies, in China there are few cultural barriers to the discussion of financial assets. Analysis revealed three types of families: 1) 'self-reliant families' who could manage elder-care themselves (4 families); 2) 'families who buy in help': that drew on external help for elder-care (3 families); and 3) 'no generation 1 families': which were those families whose G1 had passed away leaving G2 with less competition for their time (3 families).

### The self-reliant families

In these families, the gatekeeper to the family was G2 and it is this generation that provided care for their parents (G1) while delivering support to their adult children (G3) who are now parents to G4.

These families had income and wealth resources to deploy but critically, the most significant resource for this family type was the size of generation 2. Where there were several siblings in G2, they were more likely to be able to divide care between themselves though proximity of the carers to the cared for and the degree of care needed were salient.

> *We lived next to each other, and I am still living in the same neighbourhood as my mother, but my two younger brothers have moved away. I realize I have a responsibility to look after her, and not only am I the only one who lives nearby, but I am the oldest son in the family.* (G2-S1 in the Xing family)

Traditional family practice is about co-residence which seemed to work for both G1/G2 (such as in the Huo family) and G2/G3/G4 (the Li family). This living arrangement relies on financial resources that command housing space. Du [21] asserts that traditional family care

**Table 2. Families divided into types of care practice.**

| Sample family | Participant | Community | Occupation of three generations | Income of household (per year) And properties they have |
|---|---|---|---|---|
| **Elder-care by family members** | | | | |
| Li Family | G1, G2, G3. | G2-Teachers' community | G1- retired Teachers. G2-Retired Doctor, retired Professor. G3-professional technician, manager; | G1: 100,000CNY (about £11,000), 1 apartment. G2: 250,000CNY (about £27,445); 2 apartments. G3: 300,000CNY (about £32,934); |
| Hao Family | G1, G2, G3; | G2-private sector development community | G1- retired Teacher. G2-Retired teacher, Professor. G3-financial assistant, financial manager; | G1: 80,000CNY (about £8,782), 1 apartment. G2: 400,000CNY (about £43,912), 3 apartments. G3: 650,000CNY (about £71,358), 1 apartment; |
| Xing Family | G1, G2, G3. | G3- private sector development community | G1- retired Civil servant. G2-Retired workers. G3-Civil servant, editor | G1: 60,000CNY (about £6,586), 2 apartments. G2: 80,000CNY (about £8,782), live in G2-S1 parents' apartment. G3: 150,000CNY (about £16,467) live with mother-in-law. |
| Huo Family | G1, G2, G3. | G2-Teachers' community | G2-Accounting manager, nurse. G3-financial assistant | G1: 70,000CNY (about £7,684), 1 apartment. G2: 220,000CNY (about £24,152); 2 apartments. G3: 130,000CNY (about £14,271), 1 apartment; |
| **Elder-care delivered by paid care workers** | | | | |
| Ye Family | G1, G2, G3, The helper. | G2-Teachers' community | G1- retired professor, technician. G2-Retired professional technician, retired professor. G3-both assistant professor in university | G1: 100,000CNY (about £11,000), 1 apartment. G2: 180,000CNY (about £19,760); 2 apartments. G3: 80,000USD (about £61,171); 1 house in USA. |
| Zhao Family | G1, G2, G3. | G2-Teachers' community | G1- retired Teacher. G2-Retired Doctor, Professor. G3- both sales supervisors; | G1: 70,000CNY (about £7,684), 1 apartment. G2: 250,000CNY (about £27,445); 2 apartments. G3: 250,000CNY (about £27,445); |
| Wang Family | G1, G2, G3. The helper. | G2-Teachers' community | G1-Retired professional technicians. G2-Professor, editor. G3-designer | G1: 80,000CNY (about £8,782), 1 apartment. G2: 250,000CNY (about £27,445); 2 apartments. G3: 200,000CNY (about £21,956), 1 apartment; |
| **No G1 families** | | | | |
| Kong Family | G2, G3. | G3- private sector development community | G2-retired worker, salesman. G3-Housewife, design manager | G2: 80,000CNY (about £8,782). G3: 400,000CNY (about £43,912), 2 apartments; |
| Han Family | G2, G3. | G3- private sector development community | G2-private business. G3-Design manager, professional technician | G2: 80,000CNY (about £8,782); 2 apartments. G3:700,000CNY (about £76,847), 1 apartment; preparing to buy an apartment in a good school district. |
| Fu Family | G2, G3. | G2- private sector development community | G2-Retired doctor, Retired Professor. G3-both designers; | G2: 220,000CNY (about £24,152), 2 apartments. G3: 800,000CNY (about £87,825), 2 apartments; |

relies on a large family and co-residence. This is more difficult to achieve in a society where housing is increasingly dominated by market solutions. Nor is it always a welcome solution as Mr G1 in the Li family asserted:

*My three daughters are all very family oriented. They all try to persuade us to move to their new homes or somewhere near their homes. For example, my eldest daughter wanted us to move into their community; my second daughter has bought a house, and she wanted us to live with her; and my youngest daughter has purchased a flat in Qinhuangdao, which is a city famous for its health resort. We are still staying here because my wife does not want to move.*

Later in the investigation, the researcher also learnt that Mrs G1 had a sister close by which was another source of place attachment. For two of the research families a more acceptable form of co-residence may involve families living in close proximity to one another, which allows for the preservation of both care and independence.

Traditional practices that assume that the oldest son is charged with the care of his parents is challenged by the need to be more pragmatic. In the Xing family with its all-male generation 2 the eldest son is the primary care giver, but this is driven by his proximity. In the all-female generation 2, of the Li family, with the exception of G2-D3 who lives in another area, two daughters share the responsibility of visiting their parents on a weekly basis. Similarly, the G2-D2 couple of the Huo family is co-resident with G1 and as a result, they look after their elder parents even though there are two sons in this generation.

> *It (family care practices) cannot be told very precisely. It is just that kind of work, and we [my siblings] all think this is a common way to do it. My parents supported my little sister (G2-D2) in a tough period so that she would like to help [them] in their old age. I do not even think enough about that, and my little brother (G2-S3) and I also do whatever we can.* (G2-S1 in the Huo family).

Within each family, time was a precious resource that might also be demanded for childcare and so negotiation of responsibility was dynamic.

> *My oldest sister (G2-D1) and I live close to our father. We support each other, because my sister has her granddaughter to look after, so I pay more attention to our father. My sister is going to come when she has time. You know, I have a dependent boy, only 12 years old, since I was married too late, but I think that family means loving each other, which is a normal way to show love rather than say it aloud.* (G2-S4 in the Hao family).

## Families who buy in help

This group of families were approached through G2 family members. As in the first group of self-reliant families, G2 shoulders responsibility, and their tactic was to find extra support from a bought in helper. This increasingly widespread practice has been judged by some scholars [22] as evidence of declining filial piety; however as this research demonstrates motivations vary. A fine-grained exploration of the Ye family, for example, reveals the resources of people stretched to breaking point (Table 3).

**Table 3. The resource base of the Ye family.**

| Respondents | Occupation | Education | Money |
|---|---|---|---|
| **G1** Male (85), bedridden for nearly 3 months. Female (84), bedridden for 5 years. | Retired professor. Retired lead designer of a famous alcohol factory in Tianjin. | High school diploma and continued to study for a bachelor's degree | Pension: 7000 CNY (about £773) per month. 6000 CNY (about £663) per month. Own their own home |
| **G2** D1 (60) retired early in her fifties. Her husband (60) just retired in 2019 | Retired technician. her husband is a retired professor but continues to work for a private college | Bachelor's degree | Pension 5000+ CNY (more than £663) per month. 8000 CNY per month (about £884), income 5000+ (more than £663) per month. Own two properties: one to live in, one for investment |
| D2 (56) retired early | G2-D2: retired early because of disability, previously worked in G1-F's factory | High school diploma | Pension 3500 CNY (about £386) per month, and a few subsidies (additional payments for disability) |
| D3 (49) civil servant | G2-D3: civil servant. her husband and son lived in other cities because of their work and education | Bachelor's degree | Income: more than 10K CNY (more than £1105) per month |
| **G3** D1's son (33), lives in Florida with his wife | G3 couple both graduated from US university and work in Florida | PhD degree in the USA | Salary about $80K (about £61K) a year. Own their own home in USA |

The older couple are now in need of a high level of care that demands considerable time investment. The second generation is female and the three daughters in this generation have looked after their mother (Mrs. G1) for several years since the onset of her dementia—although she has now become physically frailer. In addition, Mr. G1 had a fall in January 2019, leaving him bedridden at the time of the research.

Because of his bedridden condition, it is difficult even for two family members to wash Mr. G1 and cut his hair. Normally, there is only one adult child at home to provide support, which means that the care needs of both G1 parents cannot be met at the same time. The need for more support than G2 can provide led these G2 respondents to hire a home-helper. As a couple with decent pensions, G1 have enough money to meet their own care costs. G2-D1 is the one who makes most decisions, and it was she who suggested buying in support and organised recruitment. G2-D1 holds her parents' pension cards though all her sisters have a say in how to spend their money. Knowing the traditional outlook of their father, the sisters guessed that he would not agree to external support, and this led them to say that the helper was a distant relative of G2-D1's husband. This harmless 'lie' allowed their father to feel that his wish for family-based care was followed and his dignity preserved since it was, rightly, surmised that he would be embarrassed by the offer of intimate assistance from a stranger.

The story of the Zhao family is more dynamic in that the three daughters of generation 2 utilised a number of strategies to support their parent. When G1 was over eighty, her daughters decided to hire a helper to live with her. The idea had been discussed over the five-year period since the death of their father and at the time, only the youngest daughter (G2-D3) lived in the same city as their mother. Their family practice was that G1 lived with a 24-hour live in helper and G2-D3 visited regularly. However, three years ago, their practice changed to co-residence when G2-D3 moved to a southwest city because of her business. As a result, G2-D1 persuaded G1 to live with her in Tianjin. Co-residence is a traditional family practice and provides an effective way of performing filial piety (at least in terms of time and resources). Mrs. Zhao (G2-D1) stated that she, and her husband, had moved from her hometown to Tianjin more than 20 years ago in order to be on hand for the future care of her parents.

However, only a few months later, G1 wanted to go back to her own home in Qiqihar in the northeast of China. Thus, the three daughters (G2) decided to hire another helper at home with additional support coming from each daughter in turn who would stay with their mother on a four-month rotation basis. Last year, G2-D3 also moved back to her hometown, but their family practice remained the same. G2-D2 and G2-D3 now have new-born grandchildren to look after and G2-D1 told her two sisters that she could spend more time with their mother because she has no grandchildren. Therefore, G2 is often torn between the competing interests of childcare and elder-care and needs sibling support to manage. As with the Li family, in the Zhao family the traditional family practice of multiple generations living together was only partially acceptable to Mrs G1 who has a friendship circle and values being independent in her own home. The traditional idea that older people's life-worlds should be confined to their children and grandchildren is also challenged here.

The demand for home care is rising and the market has responded by developing agencies who match helpers and families. Most of the helpers are women, many of whom may have moved from rural or less prosperous parts of China to find work. Those who hired helpers often shared 'horror' stories that revealed their relative powerlessness; thus, in one instance, a son from (G2) in the Wang family made the following unsettling comment:

*The helpers have changed several times, and they were from the same service agency. These helpers discussed my father secretly and believed he treated them too harshly. They*

*complained to their boss about my father's living patterns, and they shared details about my family's condition. Several of their gossip messages were screenshotted by their boss and sent to my brother (G2-S2). The boss aimed to ask us to persuade our father to change his lifestyle a little, which was awkward, but we (G2) do not have a choice because we need their [the helpers and service agency] support. . .* (G2-S1 in the Wang family).

## No generation one families

The third group of families were those whose oldest people (G1) had passed away. These groups were accessed through G3. The group that we have called generation two in this paper are now the oldest people in their family. As a result, Generation two's family responsibilities are now focused on their single son or daughter and their children. Although it has become more commonplace for grandparents to play a substantial caregiving function [23], each family's condition is distinct. Generation two are in the early years of their retirement and are still energetic. Their good health allows Generation three to focus on their work and in many cases, they share, or leave, the care of their little children (G4) to G2. Mrs G3 in the Han family (see Table 4) was grateful that G2 would look after her children so that she could go back to work after family leave.

*It was unfortunate that I had to leave my job because I was 8 months pregnant, so I could not focus on my job, and I did not have any inspiration for my design work. After a year's stay at home with my son, I could not wait to return to the labour market with the help of my husband and his parents. I am so glad that my father and mother-in-law can support me, unlike some of my former colleagues who were forced to be housewives.* (G3-wife in the Han family).

Unlike their own parents the future care of G2 by family members may be less certain.

In the Kong family, the perceived lack of support from G2 in the rearing of G3's young child resulted in G3 feeling less willing to invest in traditional forms of elder care. G3-D (Kong family), who was expecting her second child at the time of interview, expressed her disappointment in her parents:

*I expected that my parents would help me to take care of their grandchildren, but I was wrong. In turn, I am hesitant in my heart of hearts to support their later life. I cannot rely on my parents to take care of a baby; thus, I will pay for a professional nanny for my upcoming baby. They cannot have a say [in this] because I am spending my own money. I do not want to look after my parents, and I already regret asking them to live with us.* (G3-D in Kong family).

**Table 4. Resources of the Han family.**

| Respondents | | Occupation | Education | Money |
|---|---|---|---|---|
| G2 | Husband (63). Wife (62). | Civil servant | College | Pension: About 8K CNY (about £923) per month. Own one apartment in hometown to live in at some point in later life |
| G3 | S (36). His wife (36). | Electrical engineer. Design manager. | Bachelor's degree | Income: 350K CNY (about £40,384) per year. 400K CNY (about £46,153) per year. Own two apartments: one for G3 couple, G4 and G3-husband's parents are co-resident. One for investment. Own a car. |
| G4 | S (4) | | | |

While this is only one narrative it can be hypothesised that G3 may be more likely to weigh the value of the care and time invested in grandparent care as worthy, or not, of reciprocal care. As such responses suggest traditional expressions of filial piety that were once viewed as sacred have now been unsettled by changing attitudes across the generations.

For Generation 2 there are also new possibilities. In the Han family, Generation 2 and 3 are co-residents and as a result they can provide childcare for the little boy in their family but G2 have retained a house in their home city which they plan to return to when the responsibilities of childcare lessen.

*As for later life, we are hoping to go back to Anshan city. There are a lot of relatives in our hometown, and we should support each other. They (G3) are too busy. We cannot count on them to look after us* (G2, Han family).

Mrs G3 of the Han family however, was clear that she saw her future duty as providing care for her in-laws.

*I should no longer complain about my father and mother-in-law. Probably, my mother-in-law might be complaining to me because I never do housework at home, and something else, because I work very hard, I do not even have time to spend with my son. Of course, my career cannot be continued without their help at home. In turn, I will do whatever I can when they need support.*

They believe that their own later life care is likely to be based on reciprocal support across friendship groups and family members of their own generation.

## Discussion

The findings of this study accord with Tsutsui's [24] investigation demonstrating that standards of filial obligation are dynamic over the life-course and between generations. From the findings of this investigation, we can hypothesise that social norms in the future in China will change. Indeed, it might be the case that in the future, a young couple in Generation 3 (G3) will be supporting four older parents (G2) and their own children (G4); however, it is also likely that this couple will be living at a distance from their parents (or they may be living overseas) which will make traditional modes of caring unlikely. More disposable income adds resources to families potentially opening up more choices for caring practices, such as a personal care worker, a domestic helper or reliance on community-based day-care centres. In short, middle-class families may buy physical support and time where they are overburdened or have shrinking people resources. In the future, market-based provisions may overtake the dominant position of the family in elder-care.

Moving into a market-oriented society that emphasises personal choice presents a risk to middle class lifestyles. Indeed, middle-class lifestyles cannot easily be maintained without public welfare support because, the long-term care of older parents requires reasonable financial and/or physical costs. In this study, two types of elder care were discussed: self-reliant families (type 1) and families needing additional help (type 2). In this investigation, G1 all stated that they want to age at home (or in their original community) as long as possible. The move away from the normative expectation of traditional co-residence makes it more difficult for G2 to provide practical care, particularly when frailty occurs [25]. Employing a domestic helper allows older parents to maintain their preferences and adult children to fulfil their family duties by "borrowing others' hands."

As Li argues [20], family and culture are a lubricant that, like the human immune system, allows the country to self-regulate and self-recover, while family resources can be re-organised. This might mean that multiple siblings help each other, and the sharing of similar culturally based values tie family members together to solve challenges [20]. Ren [26], in contrast, argues that traditional rules were broken down suddenly by changes in Chinese society, but new concepts have not been normalised, which may cause new uncertainty, questions, and risk. The challenge is 'the inadequate governmental provisions for family-based and institutional-based elder-care, which is sometimes rationalised by a simplistic view of filial piety' [27]. Canda suggests that Confucian philosophy, far from demanding a fixity of approach, "emphasises the importance of timeliness and adaptation, [which means that] principles and practices should be adapted according to the real circumstances, opportunities, and historical conditions of the present and a given culture" [27]. The current adaption of filial piety in this study has shown that middle-aged people (G2) may be adjusting their expectations, so while some may provide care for grandchildren in the expectation of later-life care by G3, others may be forging different support futures. It is clear that Chinese families are questioning and critiquing traditional values that were once understood to be sacred and beyond question. However, these changes do not mean that filial obligations no longer apply, but instead a new generation of Chinese families–are looking for more nuanced solutions.

Greater socio-economic development does not mean that the family is *not valued* but rather that individuals are free to deviate from conventional family practices [28]. However, it has been argued that the performance of family care can act as a role model for younger people. The importance of caring for one's grandparents is a value that can be transmitted through family generations. Falkingham *et al*. [23], state that the generation who help older parents are also helping their children by providing them with a model of caring which can be passed onto the next generation.

## Risk and uncertainty in the new China

In parallel to those in generation 3 who are rethinking filial piety, their parents are also re-evaluating their future lives. They are less financially dependent on their adult children due to their pensions and supplementary medical insurance, but grand parenting care may result in close emotional relations [29]. Some studies argue that grandparental care is a strategy taken by older parents, to '[pay] forward for their old-age care' [30]. However, this argument does not sit well with Goh et al., whose research from Fuzhou and Singapore found that older parents are reducing their expectations of their children in that they are 'hoping for', rather than 'expecting' care from their children [29, 30].

A 2015 policy in China to allow a couple to have two children and three in 2021 has resulted in more focus on the needs of children. In addition, the proposal to raise the retirement age to increase the workforce, while also limiting the expenditure on pensions, will create more stress on Generation 2 as they struggle to meet their multiple obligations (with the added complexity of being active in the labour market). Given the growing ageing population and the paucity of support, a more flexible retirement policy is needed with care allowances for those who need to leave their employment because of elder, or childcare, responsibilities. The demand for home help gives added impetus for establishing a formalised home care sector in China with regularised standards that offers training for workers and quality assurance for families' peace of mind.

## Limitations

This study was limited by its scale and geographical focus. The decision to conduct family studies created a strong imperative to establish trust, which determined both the choice of city

and the need to build a sample from personal connections. However, Tianjin is representative of a large group of Chinese cities that have seen a rise of middle-class professionals. Moreover, the challenges and self-help solutions set out in the ten family narratives in this study can provide insights into the struggles experienced by their peers. Many of the themes in this research might be taken forward into a larger scale study by the utilisation of a sophisticated questionnaire.

## Supporting information

**S1 Appendix.**
(PDF)

**S2 Appendix. Samples of interview questions [31] (pp. 150–151).**
(DOCX)

**S1 File.**
(DOCX)

## Author Contributions

**Conceptualization:** Rose Gilroy, Andrew Law.

**Supervision:** Andrew Law.

**Writing – original draft:** Lu Wang.

**Writing – review & editing:** Rose Gilroy.

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
