## [Decision Letter · Decision Letter 0]

19 Sep 2022

PONE-D-22-15456Shifting elder care practices in Chinese middle-class familiesPLOS ONE

Dear Dr. Wang,

Thank you for submitting your manuscript to PLOS ONE. After careful consideration, we feel that it has merit but does not fully meet PLOS ONE’s publication criteria as it currently stands. Therefore, we invite you to submit a revised version of the manuscript that addresses the points raised during the review process.

Please note that we have only been able to secure a single reviewer to assess your manuscript. We are issuing a decision on your manuscript at this point to prevent further delays in the evaluation of your manuscript. Please be aware that the editor who handles your revised manuscript might find it necessary to invite additional reviewers to assess this work once the revised manuscript is submitted. However, we will aim to proceed on the basis of this single review if possible.  Please also note that the reviewer requires a major rework of almost all sections of the manuscript. This includes the discussion where the findings should be outlined in relation to existing literature. The reviewer also requires the inclusion of discussions of the limitations of the study, such as the study population, a single city, interview conditions, and bias in educational levels, and the inclusion of future implications and recommendations for research and practice. 

We look forward to receiving your revised manuscript.

Kind regards,

Alice Coles-Aldridge

Editorial Office

PLOS ONE

Journal Requirements:

Reviewers' comments:

Reviewer's Responses to Questions

**Comments to the Author**

1. Is the manuscript technically sound, and do the data support the conclusions?

Reviewer #1: Partly

2. Has the statistical analysis been performed appropriately and rigorously? 

Reviewer #1: N/A

3. Have the authors made all data underlying the findings in their manuscript fully available?

Reviewer #1: No

4. Is the manuscript presented in an intelligible fashion and written in standard English?

Reviewer #1: No

5. Review Comments to the Author

Reviewer #1: The topic of (changes in) traditional values like intergenerational family support (and more particular filial piety from the younger to the older generation) is very interesting and highly relevant in ageing societies like China but also elsewhere. The authors focus on middle class multigenerational families and found an challenging qualitative design to interview three generations in ten families. However, there are major concerns with the structure, wording and content of almost al parts of the paper:

- the Abstract should be re-written conform the requitements of the journal, including more precise information about methods, limitations and one or two implications/recommendations.

- the Introduction is very lengthy, too wordy and often hard to follow. Restrict yourselves to problem analysis, (knowledge gaps in) the most relevant literature, relevance and focus of your research question.

- Please rewrite your Methods in a more strict style, not in the present conversational way of story telling. The arguments behind your selection criteria are not substantiated and do not always seem to be selection criteria but more descriptive variables about your research population (also pleas translate Hukou). Your informal way of recruiting from personal networks has lead to a upward bias in the eductional level of your respondents, on top of your selection criterion of "middle class". I miss this as part of a paragraph about limitations in your Discussion. On page 8 you mention that "several" respondents refused audiorecording of the interviews. How many (in percentages) and how then could you make a transcription of this interview? How was all coding and data analyses (last paragraph at the end of page 8) performed, only by the first author or by more authors, how did you deal with unclearness or disconsensus about coding and intepretation of data? This is an important part of your Methods section, also to judge whether other researchers would have achieved the same results when they would follow your methods (validity!). In table 2, the relevance of the column community remains unclear and does not seem to play a role in the Findings/Results. Please provide a table or Appendix about the topic list you used to guide the interviews. How long did the interviews take, was it always one-to-one with each respondent or were other people present or did you even have small group interviews with several G-respondents at the same time? In short, describe in more precise details all steps you have undertaking in crafting your methodology. That is important to judge the quality and validity of your methods (would other researchers find the same results if the follow your methods?).

- In the present state of your paper, the Findings/Results are the better part, with an interesting typology in three types of families. However, this section will become stronger if you really concentrate on findings only, in more condensed wording and without any embedding in literature of reflective discussions, as that belongs to the following section Discussion. Tables about particular families partly overlap with table 1, and within these tables the columns Resources (do you mean Respondents?) and Occupation also overlap. Perhaps it is a suggestion to put these tables in an Appendix?

- Your Discussion needs rework. PLeas start with the main findings, reflect on the in relation to existing literature also please also include paragraphs about Limitations (about study population, in only one city, interview conditions, bias in educational levels) and Implications/Recommendations (for research and practice). I doubt the position of the last parts of "discourse of the young" and "risk and uncertainty in the new China". The first parts was not presented as a Result before, both parts could be better integrated in the Discussion (or subsection Implications), in stead of being presented as a kind of add-ons at the final end.

- The whole paper requires thorough editing by a native English corrector

6. PLOS authors have the option to publish the peer review history of their article (what does this mean?). If published, this will include your full peer review and any attached files.

Reviewer #1: **Yes: **Robbert Huijsman

---

## [Author Response · Author response to Decision Letter 0]

15 Nov 2022

Dear editor and reviewer:

Can we thank you and the reviewer for their attention to this paper that has provided us with the opportunity to strengthen the submission. We did not upload the manuscript with tracing marks, because our manuscript has a major revision and three authors all contributed on it, so we did not save the version with tracing marks. However, we did do the revision by following the reviewer’s suggestions.

To response point to point below:

1. The abstract has been rewritten in line with the advice (page 1)

2. The introduction has been radically shortened and tightened to give (we hope) a clearer line of argument. In each paragraph we have set out the situation as discussed in the literature and have posed questions that need to be addressed as a result. (pages 2-5)

3. The Methods section and new data analysis section have also been recast and are less discursive in style. They make more clear the imperative behind research approaches. (pages 5-9)

4. Relatively unfamiliar Chinese terms such as hukou have been translated for the benefit of the reader. (page 6)

5. The question of what percentage of participants who refused recording has been addressed. (page 8)

6. While the field work was undertaken by the first author all subsequent analysis was informed by discussion between the three authors which is explained. (page 9) 

7. The topic guide is provided (appendix 1).

8. There is greater clarity about the mechanics of the interviews (page 7)

9. More discursive discussion of the family types is taken into the discussion (pages 20- 25)

10. Resources means resources that families may deploy– occupation, education level, income/wealth and property ownership). To avoid confusion e.g., table 4 (page 19) we have labelled respondents and broken down the resources into elements.

11. The discussion has been reworked in line with advice and includes a section setting out the limitations.

12. The second and third authors who are native British English speakers took a greater role in reworking the paper and its language.

 Warm regards

 Lu, Rose and Andy

---

## [Decision Letter · Decision Letter 1]

30 Jan 2023

PONE-D-22-15456R1Shifting elder care practices in Chinese middle-class familiesPLOS ONE

Dear Dr. Wang,

Thank you for submitting your manuscript to PLOS ONE. After careful consideration, we feel that it has merit but does not fully meet PLOS ONE’s publication criteria as it currently stands. Therefore, we invite you to submit a revised version of the manuscript that addresses the points raised during the review process.

Please see the comments from one previous reviewer and one new reviewer below. Please consider carefully all comments made by the reviewers. One reviewer notes that some sections can be shortened - please consider this, although note that PLOS ONE does not have word limits, and as such, you may choose how you approach this.  Please also confirm when you resubmit that all surnames are mock names, and that all information has been thoroughly anonymised, and that there is no possibility of reidentification of data.

We look forward to receiving your revised manuscript.

Kind regards,

Hanna Landenmark

Staff Editor

PLOS ONE

Journal Requirements:

Reviewers' comments:

Reviewer's Responses to Questions

**Comments to the Author**

1. If the authors have adequately addressed your comments raised in a previous round of review and you feel that this manuscript is now acceptable for publication, you may indicate that here to bypass the “Comments to the Author” section, enter your conflict of interest statement in the “Confidential to Editor” section, and submit your "Accept" recommendation.

Reviewer #1: All comments have been addressed

Reviewer #2: (No Response)

2. Is the manuscript technically sound, and do the data support the conclusions?

Reviewer #1: Yes

Reviewer #2: Partly

3. Has the statistical analysis been performed appropriately and rigorously? 

Reviewer #1: N/A

Reviewer #2: N/A

4. Have the authors made all data underlying the findings in their manuscript fully available?

Reviewer #1: Yes

Reviewer #2: No

5. Is the manuscript presented in an intelligible fashion and written in standard English?

Reviewer #1: No

Reviewer #2: No

6. Review Comments to the Author

Reviewer #1: Some final check on grammar, wording, editing is still necessary (example: "eldercare" versus "elder care" (with or without space), but better seems "elderly care" and more preferably an other term like "care for older persons"; page 4: no spcae between -2 and parents); no first names in author referencing; et al. yes or not in cursive)

Reviewer #2: Thank you for the opportunity to review the revised version of Shifting elder care practices in Chinese middle-class families. I report my review by section below.

1. Abstract

- The abstract could be written more in-line with a typical research abstract (background, purpose/aims, methods, results, discussion/conclusion). As written, it is unclear the underlying methods/rigor of this study or how the points the authors refer to were identified.

- There remains grammatical errors in the abstract.

2. Introduction

- Second paragraph, second sentence. I’m not sure this sentence is asking a question.

- While interesting, the introduction can be shortened considerably. At present, it is more than 1600 words in length. In contrast, the typical medical journal manuscript is typically 3000-3500 words total.

3. Methods

- Consider shortening. This reads much more like a dissertation rather than a journal article.

- Consider using headings (e.g., sampling strategy, data analysis)

- Consider reporting methods in line with qualitative reporting guidelines (e.g., COREQ)

- The authors could consider removing several unnecessary details in the methods (i.e. gifts, small favors etc.)

- Discussion of methodological issues of the study (e.g., the discussion of why respondents preferred to not be recorded) belong in the discussion section, not methods.

- Paragraph under the image 1 belongs in discussion other than the description of IRB approval.

- More details regarding the qualitative methods are needed. How many people coded? Were transcripts professionally transcribed? What analytic method did the authors use? How did the authors code and develop themes? Did they reach data saturation? How was the interview guide developed? Refined? How were codes developed and refined? Please follow qualitative reporting guidelines as suggested above.

- Typically names of families, patients, etc. are not included in manuscripts due to concerns for HIPAA and the need to de-identify respondents. Consider changing names to abbreviations (e.g., the “H Family” or “family # 1”). Given financial information is also described in this paper (Table 4 resources of the Han family) I would strongly urge the authors to anonymize respondents.

4. Findings

- In this section only report the findings from the study. Any discussion/expansion of the findings should be included in the discussion section.

5. Discussion

- Lacking a limitations paragraph

7. PLOS authors have the option to publish the peer review history of their article (what does this mean?). If published, this will include your full peer review and any attached files.

Reviewer #1: **Yes: **Robbert Huijsman

Reviewer #2: No

---

## [Author Response · Author response to Decision Letter 1]

27 Feb 2023

Dear editor and reviews:

we upload a word document to response all comments. Please check the word file.

Thank you.

Lu, Rose, Andy

---

## [Editor Report · Decision Letter 2]

12 Mar 2023

Shifting elder care practices in Chinese middle-class families

PONE-D-22-15456R2

Dear Dr. Wang,

We’re pleased to inform you that your manuscript has been judged scientifically suitable for publication and will be formally accepted for publication once it meets all outstanding technical requirements.

Kind regards,

Robbert Huijsman, PhD

Guest Editor

PLOS ONE

Additional Editor Comments (optional):

Dear authors, PlosOne has invited me to act as Guest Academic Editor. I hereby disclose that I was reviewer #1 in the first and second round. You followed up on many of my comments in the first round, and I expressed to be happy with that in the second round. But as I was the only reviewer in the first round, it is great that PlosOne found a second reviewer in the second round. This person had some very important comments and made perfect suggestions to strengthen your paper. And gladly you did so, especially in Abstract, English language, shortening the Introduction, revising and shortening the sections for Methods and Discussion. The qualitative methods are more concise and clear now, according to the requirements of PlosOne. You assured the reviewers that you did not use actual but fictional and randomly assigned names for the families and respondents. Elements of the earlier versions are now better placed in the Discussion, as was a serious comment in both the first and second review round. Yor study is based on original research, with interesting findings about changing intergenerational attitudes and beliefs about elder-care and family care in middle class families and your analyses are followed-up by a good discussion and conclusion, with added value to the literature. Therefore, I come to the conclusion to accept the present revised version.
---

## [Editor Report · Acceptance letter]

15 Mar 2023

PONE-D-22-15456R2 

Shifting elder-care practices in Chinese middle-class families 

Dear Dr. Wang:

I'm pleased to inform you that your manuscript has been deemed suitable for publication in PLOS ONE. Congratulations! Your manuscript is now with our production department. 

Kind regards, 

on behalf of

Professor Robbert Huijsman 

Guest Editor

PLOS ONE